# Optimizing Generalized PageRank Methods for Seed-Expansion Community Detection

**Pan Li**
UIUC
panli2@illinois.edu

**Eli Chien**
UIUC
ichien3@illinois.edu

**Olgica Milenkovic**
UIUC
milenkov@illinois.edu

## Abstract

Landing probabilities (LP) of random walks (RW) over graphs encode rich information regarding graph topology. Generalized PageRanks (GPR), which represent weighted sums of LPs of RWs, utilize the discriminative power of LP features to enable many graph-based learning studies. Previous work in the area has mostly focused on evaluating suitable weights for GPRs, and only a few studies so far have attempted to derive the optimal weights of GPRs for a given application. We take a fundamental step forward in this direction by using random graph models to better our understanding of the behavior of GPRs. In this context, we provide a rigorous non-asymptotic analysis for the convergence of LPs and GPRs to their mean-field values on edge-independent random graphs. Although our theoretical results apply to many problem settings, we focus on the task of seed-expansion community detection over stochastic block models. There, we find that the predictive power of LPs decreases significantly slower than previously reported based on asymptotic findings. Given this result, we propose a new GPR, termed Inverse PR (IPR), with LP weights that increase for the initial few steps of the walks. Extensive experiments on both synthetic and real, large-scale networks illustrate the superiority of IPR compared to other GPRs for seeded community detection. [1]

## 1   Introduction

PageRank (PR), an algorithm originally proposed by Page et al. for ranking web-pages [1] has found many successful applications, including community detection [2, 3], link prediction [4] and recommender system design [5, 6]. The PR algorithm involves computing the stationary distribution of a Markov process by starting from a seed vertex and then performing either a one-step of random walk (RW) to the neighbors of the current seed or jumping to another vertex according to a predetermined probability distribution. The RW aids in capturing topological information about the graph, while the jump probabilities incorporate modeling preferences [7]. A proper selection of the RW probabilities ensures that the stationary distribution induces an ordering of the vertices that may be used to determine the "relevance" of vertices or the structure of their neighborhoods.

Despite the wide utility of PR [7, 8], recent work in the field has shifted towards investigating various generalizations of PR. Generalized PR (GPR) values enable more accurate characterizations of vertex distances and similarities, and hence lead to improved performance of various graph learning techniques [9]. GPR methods make use of arbitrarily weighted linear combinations of landing probabilities (LP) of RWs of different length, defined as follows. Given a seed vertex and another arbitrary vertex $v$ in the graph, the $k$-step LP of $v$, $x_v^{(k)}$, equals the probability that a RW starting from the seed lands at $v$ after $k$ steps; the GPR value for vertex $v$ is defined as $\sum_{k=0}^{\infty} \gamma_k x_v^{(k)}$, for some weight sequence $\{\gamma_k\}_{k \geq 0}$. Certain GPR representations, such as personalized PR (PPR)[10] or heat-kernel PR (HPR)[11], are associated with weight sequences chosen in a heuristic manner: PPR

uses traditional PR weights, $\gamma_k = (1-\alpha)\alpha^k$, for some $\alpha \in (0,1)$, and a seed set that captures locality constraints. On the other hand, HPR uses weights of the form $\gamma_k = \frac{h^k}{k!}e^{-h}$, for some $h > 0$. A question that naturally arises is what are the provably near-optimal or optimal weights for a particular graph-based learning task.

Clearly, there is no universal approach for addressing this issue, and prior work has mostly reported comparative analytic or empirical studies for selected GPRs. As an example, for community detection based on seed-expansion (SE) where the goal is to identify a densely linked component of the graph that contains a set of a priori defined seed vertices, Chung [12] proved that the HPR method produces communities with better conductance values than PPR [13]. Kloster and Gleich [14] confirmed this finding via extensive experiments over real world networks. Avron and Horesh [15] leveraged time-dependent PRs, a convolutional form of HPR and PPR [16], and showed that this new PR can outperform HPR on a number of real network datasets. Another line of work considered adaptively learning the GPR weights given access to sufficiently many both within-community and out-of-community vertex labels [17, 18]. Related studies were also conducted in other application domains such as web-ranking [8] and recommender system design [19].

Recently, Kloumann et al. [20] took a fresh look at the GPR-based seed-expansion community detection problem. They viewed LPs of different steps as features relevant to membership in the community of interest, and the GPRs as scores produced by a linear classifier that digests these features. A key observation in this setting is that the GPR weights have to be chosen with respect to the informativeness of these features. Based on the characterization of the mean-field values of the LPs over a modified stochastic block model (SBM) [21], Kloumann et al. [20] determined that PPR with a proper choice of the parameter $\alpha$ corresponds to the optimal classifier if only the first-order moments are available. Unfortunately, as the variance of the LPs was ignored, the performance of the PPR was shown to be sub-optimal even for synthetic graphs obeying the generative modeling assumptions used in [20].

We report substantial improvements of the described line of work by characterizing the non-asymptotic behavior of the LPs over random graphs. More precisely, we derive non-asymptotic conditions for the LPs to converge to their mean-field values. Our findings indicate that in the non-asymptotic setting, the discriminative power of $k$-step LPs does not necessarily deteriorate as $k$ increases; this follows since our bounds on the variance decay even faster than the distance between the means of LPs within the same and across two different communities. We leverage this finding and propose new weights that suitably increase with the length of RWs for small values of $k$. This choice differs significantly from the geometrically decaying weights used in PPR, as suggested by [20].

The reported results may also provide useful means for improving graph neural networks (GNN) [22, 23, 24] and their variants [25, 26] for vertex classification tasks. Currently, the typical numbers of layers in graph neural networks is $2-3$, as such a choice offers the best empirical performance [24, 25]. More layers may over-smooth vertex features and thus provide worse results. However, in this setting, long paths in the graphs may not be properly utilized, as our work demonstrates that these paths may have strong discriminative power for community detection. Hence a natural research direction of research regarding GNNs is to investigate how to leverage long paths over graphs without over-smoothing the vertex features. Concurrent to this work, several empirical studies were performed to address the same problem. The work in [27, 28] used a decoupling non-linear transformation of features and PR propagation over graphs, while [29] used GNNs over graphs that are transformed based on GPRs.

Our contributions are multifold. We derive the first non-asymptotic bound of the distance between LP vectors to their mean-field values over random graphs. This bound allows us to better our understanding of a class of GPR-based community detection approaches. For example, it explains why PPR with a parameter $\alpha \simeq 1$ often achieves good community detection performance [30] and why HPR statistically outperforms PPR for community detection, which matches the combinatorial demonstration proposed previously [12]. Second, we describe the first non-asymptotic characterization of GPRs with respect to their mean-field values over edge-independent random graphs. The obtained results improve the previous analysis of standard PR methods [31, 32] as one needs fewer modeling assumptions and arrives at more general conclusions. Third, we introduce a new PR-type classifier for SE community detection, termed inverse PR (IPR). IPR carefully selects the weights for the first several steps of the RW by taking into account the variance of the LPs, and offers significantly improved SE community detection performance compared to canonical PR diffusions (such as HPR

and PPR) over SBMs. Fourth, we present extensive experiments for detecting seeded communities in *real large-scale networks* using IPR. Although real world networks do not share the properties of SBMs used in our analysis, IPR still significantly outperforms both HPR and PPR for networks with non-overlapping communities and offers performance improvement over two examined networks with overlapping community structures.

## 2 Preliminaries

We start by formally introducing LPs, GPR methods, random graphs and other relevant notions.

**Generalized PageRank.** Consider an undirected graph $G = (V, E)$ with $|V| = n$. Let $A$ be the adjacency matrix, and let $D$ be the diagonal degree matrix of $G$. The RW matrix of $G$ equals $W = AD^{-1}$. Let $\{\lambda_i\}_{i \in [n]}$ be the eigenvalues of $W$ ordered as $1 = \lambda_1 \geq \lambda_2 \geq ... \geq \lambda_n \geq -1$. Furthermore, let $d_{\min}$ and $d_{\max}$ stand for the minimum and maximum degree of vertices in $V$, respectively. A distribution over the vertex set $V$ is a mapping $x : V \to \mathbb{R}_{[0,1]}$ such that $\sum_{v \in V} x_v = 1$, with $x_v$ denoting the probability of vertex $v$. Given an initial distribution $x^{(0)}$, the $k$-**step LPs** equal $x^{(k)} = W^k x^{(0)}$. The **GPRs** are parameterized by a sequence of nonnegative weights $\gamma = \{\gamma_k\}_{k \geq 0}$ and an initial potential $x^{(0)}$, $pr(\gamma, x^{(0)}) = \sum_{k=0}^{\infty} \gamma_k x^{(k)} = \sum_{k=0}^{\infty} \gamma_k W^k x^{(0)}$. For an in-depth discussion of PageRank methods, the interested reader is referred to the review [7]. In some practical GPR settings, the bias caused by varying degrees is compensated for through degree normalization [33]. The $k$-**step degree-normalized LPs (DNLP)** are defined as $z^{(k)} = \left(\sum_{v \in V} d_v\right) D^{-1} x^{(k)}$.

**Random graphs.** Throughout the paper, we assume that the graph $G$ is sampled according to a probability distribution $P$. The mean-field of $G$ with respect to $P$ is an undirected graph $\bar{G}$ with adjacency matrix $\bar{A} = \mathbb{E}[A]$, where the expectation is taken with respect to $P$. Similarly, the mean-field degree matrix is defined as $\bar{D} = \mathbb{E}[D]$ and mean-field random walk matrix as $\bar{W} = \bar{A}\bar{D}^{-1}$. The mean-field GPR reads as $\bar{pr}(\gamma, x^{(0)}) = \sum_{k=0}^{\infty} \gamma_k \bar{x}^{(k)} = \sum_{k=0}^{\infty} \gamma_k \bar{W}^k x^{(0)}$. We also use the notation $\bar{z}^{(k)}, \bar{d}_{\min}$, and $\bar{d}_{\max}$ for the mean-field counterparts of $z^{(k)}, d_{\min}$, and $d_{\max}$, respectively.

For the convergence analysis, we consider a sequence of random graphs $\{G^{(n)}\}_{n \geq 0}$ with increasing size $n$, sampled using a corresponding sequence of distributions $\{P^{(n)}\}_{n \geq 0}$. For a given initial distribution $\{x^{(0,n)}\}_{n \geq 0}$ and weights $\{\gamma^{(n)}\}_{n \geq 0}$, we aim to analyze the conditions under which the LPs $x^{(k,n)}$ and GRPs $pr(\gamma^{(n)}, x^{(0,n)})$ converge to their corresponding mean-field counterparts $\bar{x}^{(k,n)}$ and $\bar{pr}(\gamma^{(n)}, x^{(0,n)})$, respectively. We say that an event occurs *with high probability* if it has probability at least $1 - n^{-c}$, for some constant $c$. If no confusion arises, we omit $n$ from the subscript. We also use $\|x\|_p = \left(\sum_{v \in V} |x_v|^p\right)^{\frac{1}{p}}$ to measure the distance between LPs.

**Edge-independent random graphs and SBMs.** Edge-independent models include a wide range of random graphs, such as Erdős-Rényi graphs [34], Chung-Lu models [35], stochastic block models (SBM) [21] and degree corrected SBMs [36]. In an edge-independent model, for each pair of vertices $u, v \in V$, an edge $uv$ is drawn according to the Bernoulli distribution with parameter $p_{uv} \in [0, 1]$ and the draws for different edges are performed independently. Hence, $\mathbb{E}[A_{uv}] = p_{uv}$, and $A_{uv}, A_{u'v'}$ are independent if $uv, u'v'$ are different unordered pairs.

Some of our subsequent discussion focuses on two-block SBMs. In this setting, we let $C_1, C_0 \subset V$ denote the two blocks, such that $|C_1| = n_1$ and $|C_0| = n_0$. For any pair of vertices from the same block $u, v \in C_i$, we set $p_{uv} = p_i$, for some $p_i \in (0, 1)$, $i \in \{0, 1\}$. Note that we allow self loops, i.e. we allow $u = v$, which makes for simpler notation without changing our conclusions. For pairs $uv$ such that $u \in C_1$ and $v \in C_0$, we set $p_{uv} = q$, for some $q \in (0, 1)$. A two-block SBM in this setting is parameterized by $(n_1, p_1, n_0, p_0, q)$.

## 3 Mean-field Convergence Analysis of LPs and GPRs

In what follows, we characterize the conditions under which $x^{(k)}$ and $pr(\gamma, x^{(0)})$ converge to their mean-field counterparts $\bar{x}^{(k)}$ and $\bar{pr}(\gamma, x^{(0)})$, respectively. The derived results enable a subsequent analysis of the variance of LPs over SBM, as outlined in the sections to follow (all proofs are postponed to Section B of the Supplement). Note that since GPRs are linear combinations of LPs,

the convergence properties of $x^{(k)}$ may be used to analyze the convergence properties of $pr(\gamma, x^{(0)})$. More specifically, given a sequence of graphs of increasing sizes, and $G^{(n)} \sim P^{(n)}$, the first question of interest is to derive non-asymptotic bounds for $\|x^{(k)} - \bar{x}^{(k)}\|_1$, as both $x^{(k)}, \bar{x}^{(k)}$ have unit $\ell_1$-norms[2]. The following lemma shows that under certain conditions, one cannot expect convergence in the $\ell_1$ norm for arbitrary values of $k$.

**Lemma 3.1.** If there exists a vertex $v$ that may depend on $n$ such that $\bar{d}_v = \omega(1)$ and $\bar{d}_v \leq (1-\epsilon)n$, for some $\epsilon > 0$, setting $x^{(0)} = 1_v$ gives $\lim_{n \to \infty} \mathbb{P}\left[\|x^{(1)} - \bar{x}^{(1)}\|_1 \geq \epsilon\right] = 1$.

Consequently, we start with an upper bound for $\|x^{(k)} - \bar{x}^{(k)}\|_2$. Then, we provide conditions that ensure that $\|x^{(k)} - \bar{x}^{(k)}\|_2 = o(\sqrt{\frac{1}{n}})$. As $\|x^{(k)} - \bar{x}^{(k)}\|_1 \leq \sqrt{n}\|x^{(k)} - \bar{x}^{(k)}\|_2$, we subsequently arrive at necessary conditions for convergence in the $\ell_1$-norm. The novelty of our proof technique is to use mixing results for RWs to characterize the upper bound for the convergence of landing probabilities for each $k$. The results establish uniform convergence of GPRs as long as $\sum_k \gamma_k < \infty$. This finding improves the results in [31, 32, 37] for GPRs with weights $\gamma_k$ that scale as $O(c^k)$, where $c \in (0,1)$ denotes the damping factor.

Our first relevant results are non-asymptotic bounds for the $\ell_2$-distance between LPs and their mean-field values. The obtained bounds lead to non-asymptotic bounds for the $\ell_2$-distance between GPRs and their mean-field values, described in Lemma 3.2. Lemma 3.2 is then used to derive conditions for convergence of LPs and GPRs in the $\ell_1$-distance, summarized in Theorems 3.3 and 3.4, respectively.

**Lemma 3.2.** Let $\bar{\lambda} = \max\{|\bar{\lambda}_2|, |\bar{\lambda}_n|\}$. Suppose that $\bar{d}_{\min} = \omega(\log n)$. Then, with high probability, and for some constants $C_1$, $C_2$, $C_3$ that do not depend on $n$ or $k$, one has

$$\frac{\|x^{(k)} - \bar{x}^{(k)}\|_2}{\|x^{(0)}\|_2} \leq C_1 \sqrt{\frac{\log n}{n\bar{d}_{\min}}} \frac{1}{\|x^{(0)}\|_2} + C_2 k \left(\bar{\lambda} + C_3\sqrt{\frac{\log n}{\bar{d}_{\min}}}\right)^{k-1} \sqrt{\frac{\bar{d}_{\max}\log n}{\bar{d}_{\min}^2}}. \quad (1)$$

Moreover, let $g(\gamma, \bar{\lambda}, \bar{d}_{\min}) = \sum_{k \geq 1} \gamma_k k \left(\bar{\lambda} + C_3\sqrt{\frac{\log n}{\bar{d}_{\min}}}\right)^{k-1}$. Then,

$$\frac{\|pr(\gamma, x^{(0)}) - \bar{pr}(\gamma, x^{(0)})\|_2}{\|x^{(0)}\|_2} \leq C_1\sqrt{\frac{\log n}{n\bar{d}_{\min}}} \frac{1}{\|x^{(0)}\|_2} + C_2 g(\gamma, \bar{\lambda}, \bar{d}_{\min})\sqrt{\frac{\bar{d}_{\max}\log n}{\bar{d}_{\min}^2}}. \quad (2)$$

Lemma 3.2 allows us to establish the following conditions for $\ell_1-$convergence of the LPs.

**Theorem 3.3.** 1) If $\|x^{(0)}\|_2 = O(\frac{1}{\sqrt{n}})$ and $\frac{\bar{d}_{\max}\log n}{\bar{d}_{\min}^2} = o(1)$, then for any sequence $\{k^{(n)}\}_{n \geq 0}$, $\|x^{(k^{(n)})} - \bar{x}^{(k^{(n)})}\|_1 = o(1)$, $w.h.p.$; 2) If $\bar{d}_{\min} = \omega(\log n)$ and $\bar{\lambda} < 1 - c$, for some $c > 0$ and $n \geq n_0$ such that $\frac{c}{3} > C_4\sqrt{\frac{\log n}{\bar{d}_{\min}}}$ where $n_0, C_4$ are constants, then for any $x^{(0)}$ and sequence $\{k^{(n)}\}_{n \geq n_0}$ that satisfies $k^{(n)} \geq (\log n + \log \frac{\bar{d}_{\max}}{\bar{d}_{\min}})/c$, we have $\|x^{(k^{(n)})} - \bar{x}^{(k^{(n)})}\|_1 = o(1)$, $w.h.p.$

Theorem 3.3 asserts that either broadly spreading the seeds, i.e., $\|x^{(0)}\|_2 = O(\frac{1}{\sqrt{n}})$, or allowing for the RW to progress until the mixing time, i.e., $k^{(n)} \geq (\log n + \log \frac{\bar{d}_{\max}}{\bar{d}_{\min}})/c$, ensures that the LPs converge in $\ell_1$-distance. One also has the following corresponding convergence result for GPRs.

**Theorem 3.4.** 1) If $\|x^{(0)}\|_2 = O(\frac{1}{\sqrt{n}})$, $\frac{\bar{d}_{\max}\log n}{\bar{d}_{\min}^2} = o(1)$, and $\bar{\lambda} < 1-c$ for some $c > 0$, then for any weight sequence $\{\gamma^{(n)}\}_{n \geq 0}$ such that $\sum_k \gamma_k^{(n)} < \infty$, one has $\|pr(\gamma^{(n)}, x^{(0)}) - \bar{pr}(\gamma^{(n)}, x^{(0)})\|_1 = o(1)$, $w.h.p.$ 2) If $\gamma_0^{(n)}/\sum_k \gamma_k^{(n)} \geq C_5 > 0$ for some constant $C_5$, $\bar{\lambda} < 1 - c$ for some $c > 0$, and $\frac{\bar{d}_{\max}\log n}{\bar{d}_{\min}^2} = o(1)$, then for any $x^{(0)}$ one has $\frac{\|pr(\gamma^{(n)}, x^{(0)}) - \bar{pr}(\gamma^{(n)}, x^{(0)})\|_2}{\|\bar{pr}(\gamma^{(n)}, x^{(0)})\|_2} = o(1)$ $w.h.p.$ 3) If $\bar{d}_{\min} = \omega(\log n)$ and $g(\gamma^{(n)}, \bar{\lambda}^{(n)}) = \sum_{k \geq 1} \gamma_k^{(n)} k (\bar{\lambda}^{(n)} + C_6)^{k-1} = O(\sqrt{\frac{\bar{d}_{\min}}{n\bar{d}_{\max}}})$ for some constant $C_6 > 0$, then for any $x^{(0)}$ one has $\|pr(\gamma^{(n)}, x^{(0)}) - \bar{pr}(\gamma^{(n)}, x^{(0)})\|_1 = o(1)$ $w.h.p.$

**Remarks pertaining to Theorem 3.4**: The result in 1) requires weaker conditions than Proposition 1 in [31] for the standard PR: we disposed of the constraint $\bar{\lambda} = o(1)$ and bounded $\bar{d}_{\max}/\bar{d}_{\min}$. As a result, GPR converges in $\ell_1$-norm as long as the initial seeds are sufficiently spread and $\frac{\bar{d}_{\max} \log n}{\bar{d}_{\min}^2} = o(1)$. The result in 2) implies that for fixed weights that do not depend on $n$, both the standard PR and HPR have guaranteed convergence in the relative $\ell_2$-distance. This generalizes Theorem 1 in [32] stated for the standard PR on SBMs. The result in 3) implies that as long as the weights $\gamma_k^{(n)}$ appropriately depend on $n$, convergence in the $\ell_1$-norm is guaranteed (e.g., for HPR with $h > (\ln n + \ln \frac{\bar{d}_{\max}}{\bar{d}_{\min}})/(2 - 2\bar{\lambda}))$.

The following lemma uses the same proof techniques as Lemma 3.2 to provide an upper bound on the distance between the DNLPs $z^{(k)}$ and $\bar{z}^{(k)}$, which we find useful in what follows. The result essentially removes the dependence on the degrees in the first term of the right hand side of (1).

**Lemma 3.5.** Suppose that the conditions of Lemma 3.2 are satisfied. Then, one has

$$\frac{\|z^{(k)} - \bar{z}^{(k)}\|_2}{\|z^{(0)}\|_2} \leq C_1 k \left( \bar{\lambda} + C_2 \sqrt{\frac{\log n}{\bar{d}_{\min}}} \right)^{k-1} \sqrt{\frac{\bar{d}_{\max} \log n}{\bar{d}_{\min}^2}} \ w.h.p.$$

# 4 GPR-Based SE Community Detection

One important application of PRs is in SE community detection: For each vertex $v$, the LPs $\{x_v^{(k)}\}_{k \geq 0}$ may be viewed as features and the GPR as a score used to predict the community membership of $v$ by comparing it with some threshold [20]. Kloumann et al. [20] investigated mean-field LPs, i.e., $\{\bar{x}_v^{(k)}\}_{k \geq 0}$, and showed that under certain symmetry conditions, PPR with $\alpha = \bar{\lambda}_2$ corresponds to an optimal classifier for one block in an SBM, given only the first-order moment information. However, accompanying simulations revealed that PPR underperforms with respect to classification accuracy. As a result, Fisher's linear discriminant [38] was used instead [20] by *empirically* leveraging information about the second-order moments of the LPs, and was showed to have a performance almost matching that of belief propagation, a statistically optimal method for SBMs [39, 40, 41].

In what follows, we rigorously derive an explicit formula for a variant of Fisher's linear discriminant by taking into account the individual variances of the features while neglecting their correlations. This explicit formula provides new insight into the behavior of GPR methods for SE community detection in SBMs and will be later generalized to handle real world networks (see Section 5).

## 4.1 Pseudo Fisher's Linear Discriminant

Suppose that the mean vectors and covariance matrices of the features from two classes $C_0$, $C_1$ are equal to $(\mu_0, \Sigma_0)$ and $(\mu_1, \Sigma_1)$, respectively. For simplicity, assume that the covariance matrices are identical, i.e., $\Sigma_0 = \Sigma_1 = \Sigma$. The Fisher's linear discriminant depends on the first two moments (mean and variance) of the features [38], and may be written as $F(x) = [\Sigma^{-1}(\mu_1 - \mu_0)]^T x$. The label of a data point $x$ is determined by comparing $F(x)$ with a threshold.

Neglecting the differences in the second order moments by assuming that $\Sigma = \sigma^2 I$, Fisher's linear discriminant reduces to $G(x) = (\mu_1 - \mu_0)^T x$, which induces a decision boundary that is orthogonal to the difference between the means of the two classes; $G(x)$ is optimal under the assumptions that only the first-order moments $\mu_1$ and $\mu_0$ are available.

The two linear discriminants have different practical advantages and disadvantages in practice. On the one hand, $\Sigma$ can differ significantly from $\sigma^2 I$, in which case $G(x)$ performs much worse than $F(x)$. On the other hand, estimating the covariance matrix $\Sigma$ is nontrivial, and hence $F(x)$ may not be available in a closed form. One possible choice to mitigate the above drawbacks is to use what we call the *pseudo Fisher's linear discriminant*,

$$SF(x) = [\text{diag}(\Sigma)^{-1}(\mu_1 - \mu_0)]^T x, \tag{3}$$

where $\text{diag}(\Sigma)$ is the diagonal matrix of $\Sigma$; $\text{diag}(\Sigma)$ preserves the information about variances, but neglects the correlations between the terms in $x$. This discriminant essentially allows each feature to contribute equally to the final score. More precisely, given a feature of a vertex $v$, say $x_v^{(k)}$, its

corresponding weight according to $SF(\cdot)$ equals $\frac{\mu_1^{(k)}-\mu_0^{(k)}}{(\sigma^{(k)})^2}$, where $(\sigma^{(k)})^2$ denotes the variance of the feature (i.e., the $k$-th component in the diagonal of $\Sigma$). Note that this weight may be rewritten as $\frac{\mu_1^{(k)}-\mu_0^{(k)}}{\sigma^{(k)}} \times \frac{1}{\sigma^{(k)}}$; the first term is a frequently-used metric for characterizing the predictiveness of a feature, called the *effect size* [42], while the second term is a normalization term that positions all features on the same scale.

Next, we derive an expression for $SF(x)$ pertinent to SE community detection, following the setting proposed for Fisher's linear discriminant in [20]. To model the community to be detected with seeds and the out-of-community portion of a graph respectively, we focus on two-block SBMs with parameters $(n_1, p_1, n_0, p_0, q)$, and characterize both the means $\mu_1$, $\mu_0$ and the variances $\mathrm{diag}(\Sigma)$. Note that for notational simplicity, we first work with DNLPs $\{z_v^{(k)}\}_{k\geq 0}$ as the features of choice, as they can remove degree-induced noise; the results for LPs $\{x_v^{(k)}\}_{k\geq 0}$ are only stated briefly.

## 4.2 $SF(\cdot)$ **Weights and the Inverse PageRank**

**Characterization of the means.** Consider a two-block SBM with parameters $(n_1, p_1, n_0, p_0, q)$. Without loss of generality, assume that the seed lies in block $C_1$. Due to the block-wise symmetry of $\bar{A}$, for a fixed $k \geq 1$, the DNLP $\bar{z}_v^{(k)}$ is a constant for all $v \in C_i$ within the same community $C_i$, $i \in \{0,1\}$. Consequently, the mean of the $k$th DNLP (feature) of block $C_i$ is set to $\mu_i^{(k)} = \bar{z}_v^{(k)}$, $v \in C_i, i \in \{0,1\}$. Note that $\bar{z}_v^{(k)}$ does not match the traditional definition of the expectation $\mathbb{E}(z_v^{(k)})$, although the two definitions are consistent when $n_1, n_0 \to \infty$ due to Lemma 3.5.

Choosing the initial seed set to lie within one single community, e.g. $\sum_{v \in C_1} x_v^{(0)} = 1$, and using some algebraic manipulations (see Section C of the Supplement), we obtain

$$\mu_1^{(k)} - \mu_0^{(k)} = c\bar{\lambda}_2^k, \quad c = \frac{1 - \bar{\lambda}_2}{n_1(n_1p_1 + n_0q)}. \tag{4}$$

Recall that $\bar{\lambda}_2$ stands for the second largest eigenvalue of the mean-field random walk matrix $\bar{W}$. The result in (4) shows that the distance between the means of the DNLPs of the two classes decays with $k$ at a rate $\bar{\lambda}_2$. This result is similar to its counterpart in [20] for LPs $\{x_v^{(k)}\}_{k\geq 0}$, but the results in [20] additionally requires $\bar{d}_v = \bar{d}_u$ for vertices $u$ and $v$ belonging to different blocks. By only using the difference $\mu_1^{(k)} - \mu_0^{(k)}$ without the variance, the authors of [20] proposed to use the discriminant $G(x) = (\mu_1 - \mu_0)^T x$, which corresponds to PPR with $\alpha = \bar{\lambda}_2$.

**Characterization of the variances.** Characterizing the variance of each feature is significantly harder than characterizing the means. Nevertheless, the results reported in Lemma 3.5 and Lemma 3.2 allow us to determine both $\mathbb{E}(z_v^{(k)} - \bar{z}_v^{(k)})^2$ and $\mathbb{E}(x_v^{(k)} - \bar{x}_v^{(k)})^2$. Let us consider $z_v^{(k)}$ first. Lemma 3.5 implies that with high probability, $\|z^{(k)} - \bar{z}^{(k)}\|_2 \leq k\left(\bar{\lambda} + o(1)\right)^{k-1}$ for all $k$. Figure 1 (Left) depicts the empirical value of $\mathbb{E}[\|z^{(k)} - \bar{z}^{(k)}\|_2^2]$ for a given set of parameter choices. As it may be seen, the expectation decays with a rate roughly equal to $\lambda_2^{2k}$, where $\lambda_2$ is the second largest eigenvalue of the RW matrix $W$. With regards to $x_v^{(k)}$, Lemma 3.2 establishes that $\|x^{(k)} - \bar{x}^{(k)}\|_2$ is upper bounded by $k\left(\bar{\lambda} + o(1)\right)^{k-1}$; for large $k$, the norm is dominated by the first term in (1), induced by the variance of the degrees. Figure 1 (Left) plots the empirical values of $\mathbb{E}[\|x^{(k)} - \bar{x}^{(k)}\|_2^2]$ to support this finding.

Combining the characterizations of the means and variances, we arrive at the following conclusions.

**Normalized degree case.** Although the expression established in (4) reveals that the distance between the means of the landing probabilities decays as $\bar{\lambda}_2^k$, the corresponding standard deviation $\sigma^{(k)} \propto \mathbb{E}[\|z^{(k)} - \bar{z}^{(k)}\|_2]$ also roughly decays as $\lambda_2^k$. Hence, for the classifier $SF(\cdot)$, the appropriate weights are $\gamma_k = \frac{\mu_1^{(k)}-\mu_0^{(k)}}{(\sigma^{(k)})^2} \sim \bar{\lambda}_2^k/\lambda_2^{2k} = \left(\bar{\lambda}_2/\lambda_2\right)^k \lambda_2^{-k}$. The first term $\left(\bar{\lambda}_2/\lambda_2\right)^k$ in the product weighs different DNLPs according to their effect sizes [42]. Since $\lambda_2 \to \bar{\lambda}_2$ as $n \to \infty$, the ratio may decay very slowly as $k$ increases. As shown in the Figure 1 (Right), the classification error rate based on a one-step DNLP remains largely unchanged as $k$ increases to some value exceeding the mixing time. The second term in the product, $\lambda_2^{-k}$, may be viewed as a factor that balances the scale of all DNLPs. Due to the observed variance, DNLPs with large $k$ should be assigned weights much larger than those used in $G(x)$, i.e., $\gamma_k = \mu_1^{(k)} - \mu_0^{(k)} = \bar{\lambda}_2^k$ as suggested in [20].

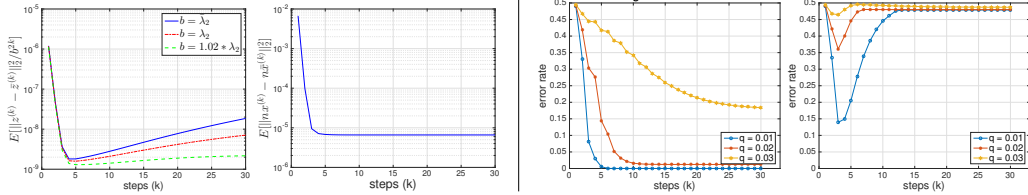

Figure 1: Left: Empirical results illustrating the decay rate of the variances $\|z^{(k)} - \bar{z}^{(k)}\|_2^2$ and $\|x^{(k)} - \bar{x}^{(k)}\|_2^2$, for an SBM with parameters $(500, 0.2, 500, 0.2, 0.05)$, averaged over 1000 tests. With high-probability, $\lambda_2$ slightly exceeds the corresponding mean-field value $\bar{\lambda}_2$ [43]; Right: Classification errors based on single-step DNLPs or LPs for SBMs with parameters $(500, 0.05, 500, 0.05, q)$, $q \in \{0.01, 0.02, 0.03\}$.

**The unnormalized degree case.** The standard deviation $\sigma^{(k)} \propto \mathbb{E}[\|x^{(k)} - \bar{x}^{(k)}\|_2]$ roughly scales as $\phi + \lambda_2^k$, where $\phi$ captures the noise introduced by the degrees. Typically, for a small number of steps $k$, the noise introduced by degree variation is small compared to the total noise (See Figure 1 (Left)). Hence, we may assume that $\phi < \lambda_2^0 = 1$. The classifier $SF(\cdot)$ suggests using the weights $\gamma_k = \frac{\mu_1^{(k)} - \mu_0^{(k)}}{(\sigma^{(k)})^2} \sim \bar{\lambda}_2^k/(\phi + \lambda_2^k) \times (\phi + \lambda_2^k)^{-1}$, where $\bar{\lambda}_2^k/(\phi + \lambda_2^k)$ represents the effect size of the $k$-th LP. This result is confirmed by simulations: In Figure 1 (Right), the classification error rate based on a one-step LP decreases for small $k$ and increase after $k$ exceeds the mixing time. Moreover, by recalling that $x_v^{(k)} \to d_v / \sum_v d_v$ as $k \to \infty$, one can confirm that the degree-based noise deteriorates the classification accuracy.

**Inverse PR.** As already observed, for finite $n$ and with high probability, $\lambda_2$ only slightly exceeds $\bar{\lambda}_2$. Moreover, for SBMs with unknown parameters or for real world networks, $\bar{\lambda}_2$ may not be well-defined, or it may be hard to compute numerically. Hence, in practice, one may need to use the heuristic value $\bar{\lambda}_2 = \lambda_2 = \theta$, where $\theta$ is a parameter to be tuned. In this case, $SF(\cdot)$ with degree normalization is associated with the weights $\gamma_k = \theta^{-k}$, while $SF(\cdot)$ without degree normalization is associated with the weights $\gamma_k = \theta^k/(\phi + \theta^k)^2$. When $k$ is small, $\gamma_k$ roughly increases as $\theta^{-k}$; we term a PR with this choice of weights as the *Inverse PR* (IPR). Note that IPR with degree normalization may not converge in practice, and LP information may be estimated only for a limited number of $k$ steps. Our experiments on real world networks reveal that a good choice for the maximum value of $k$ is $4 - 5$ times the maximal length of the shortest paths from all unlabeled vertices to the set of seeds.

**Other insights.** Note that IPR resembles HPR when $k$ is small and $\gamma_k$ increases, as it dampens the contributions of the first several steps of the RW. This result also agrees with the combinatorial analysis in [12] that advocates the use of HPR for community detection. Note that IPR with degree normalization has monotonically increasing weights, which reflects the fact that community information is preserved even for large-step LPs. To some extent, this result can be viewed as a theoretical justification for the empirical fact that PPR is often used with $\alpha \simeq 1$ to achieve good community detection performance [30].

## 5 Experiments

We evaluate the performance of the IPR method over synthetic and large-scale real world networks.

**Datasets.** The network data used for evaluation may be classified into three categories. The first category contains networks sampled from two-block SBMs that satisfy the assumptions used to derive our theoretical results. The second category includes three real world networks, Citeseer [44], Cora [45] and PubMed [46], all frequently used to evaluate community detection algorithms [47, 48]. These networks comprise several non-overlapping communities, and may be roughly modeled as SBMs. The third category includes the Amazon (product) network and the DBLP (collaboration) network from the Stanford Network Analysis Project [49]. These networks contain thousands of overlapping communities, and their topologies differ significantly from SBMs (see Table 2 in the Supplement for more details). For synthetic graphs, we use single-vertex seed-sets; for real world graphs, we select 20 seeds uniformly at random from the community of interest.

**Comparison of the methods.** We compare the proposed IPRs with PPR and HPR methods, both widely used for SE community detection [14, 50]. Methods that rely on training the weights were not

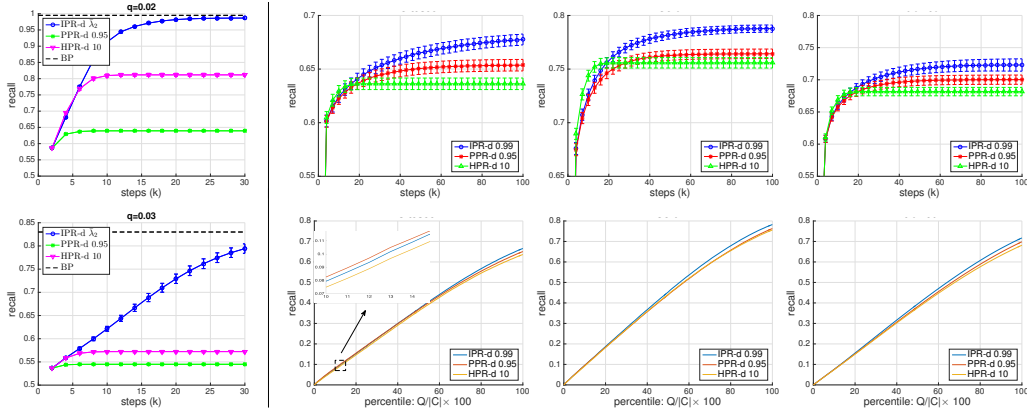

Figure 2: (Left): Recalls (mean $\pm$ std) for different PRs over SBMs with parameters $(500, 0.05, 500, 0.05, q)$, $q \in \{0.02, 0.03\}$; (Right): Results over the Citeseer, Cora and PubMed networks (from left to right). First line: Recalls (mean $\pm$ std) of different PRs vs steps. The second line: Averaged recalls of different PRs for the top-$Q$ vertices, obtained by accumulating the LPs with $k \le 50$.

considered as they require outside-community vertex labels. For all three approaches, the default choice is degree-normalization, indicated by the suffix "-d". For synthetic networks, the parameter $\theta$ in IPR is set to $\bar{\lambda}_2 = \frac{0.05-q}{0.05+q}$, following the recommendations of Section 4.2. For real world networks, we avoid computing $\lambda_2$ exactly and set $\theta \in \{0.99, 0.95, 0.90\}$. The parameters of the PPR and HPR are chosen to satisfy $\alpha \in \{0.9, 0.95\}$ and $h \in \{5, 10\}$ and to offer the best performance, as suggested in [50, 51, 14]. The results for all PRs are obtained by accumulating the values over the first $k$ steps; the choice for $k$ is specified for each network individually.

**Evaluation metric.** We adopt a metric similar to the one used in [50]. There, one is given a graph, a hidden community $\mathcal{C}$ to detect, and a vertex budget $Q$. For a potential ordering of the vertices, obtained via some GPR method, the top-$Q$ set of vertices represents the predicted community $\mathcal{P}$. The evaluation metric used is $|\mathcal{P} \cap \mathcal{C}|/|\mathcal{C}|$. By default, $Q = |\mathcal{C}|$, if not specified otherwise. Other metrics, such as the Normalized Mutual Information and the F-score may be used instead, but since they require additional parameters to determine the GPR classification threshold, the results may not allow for simple and fair comparisons. For SBMs, we independently generated 1000 networks for every set of parameters. For each network, the results are summarized based on 1000 independently chosen seed sets for each community-network pair and then averaged over over all communities.

## 5.1 Performance Evaluation

**Synthetic graphs.** In synthetic networks, all three PRs with degree normalization perform significantly better than their unnormalized degree counterparts. Thus, we only present results for the first class of methods in Figure 2 (Left). As predicted in Section 4.2, IPR-d offers substantially better detection performance than either PPR-d and HPR-d, and is close in quality to belief propagation (BP). Note that the recall of IPR-d keeps increasing with the number of steps. This means that even for large values of $k$, the landing probabilities remain predictive of the community structures, and decreasing the weights with $k$ as in HPR and PPR is not appropriate for these synthetic graphs. The classifier $G(x)$, i.e., a PPR with parameters $\frac{p-q}{p+q}$ suggested by [20], has worse performance than the PPR method with parameter $0.95$ and is hence not depicted.

**Citeseer, Cora and PubMed.** Here as well, PRs with degree normalization perform better than PRs without degree normalization. Hence, we only display the results obtained with degree normalization. The first line of Figure 2 (Right) shows that IPR-d $0.99$ significantly outperforms both PPR-d and HPR-d for all three networks. Moreover, the performance of IPR-d $0.99$ improves with increasing $k$, once again establishing that LPs for large $k$ are still predictive. The results for IPR-d $0.90$, $0.95$ and a related discussion are postponed to Section A.1 in the Supplement.

The second line of Figure 2 (Right) illustrates the rankings of vertices within the predicted community given the first 50 steps of the RW. Note that only for the Citeseer network does PPR provide a better ranking of vertices in the community for small $Q$; for the other two networks, IPR outperforms PPR and HPR on the whole ranking of vertices.

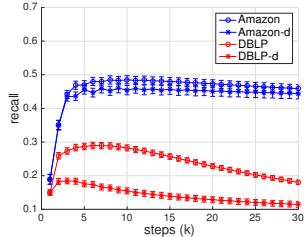

Figure 3: Recalls based on one-step LPs and one-step DNLPs.

| Steps $k$ | 5 | 10 | 15 | 20 | 5 | 10 | 15 | 20 |
|---|---|---|---|---|---|---|---|---|
| | Amazon (std: $\pm 0.12$) | | | | DBLP (std: $\pm 0.09$) | | | |
| IPR0.99 | 46.63 | 48.03 | **48.43** | **48.53** | 27.58 | 28.78 | **29.18** | **29.27** |
| IPR0.95 | 46.64 | 48.04 | **48.44** | **48.53** | 27.60 | 28.94 | **29.20** | **29.28** |
| IPR0.90 | 46.67 | 48.08 | **48.45** | **48.53** | 27.64 | **29.14** | **29.26** | **29.32** |
| PPR | 46.57 | 47.92 | 48.30 | 48.43 | 27.46 | 28.49 | 28.90 | 29.06 |
| HPR | **47.20** | **48.36** | **48.54** | **48.55** | **28.24** | 28.80 | 28.85 | 28.85 |

Table 1: Recalls (mean $\pm$ std) for different PRs over the Amazon and the DBLP networks. The boldfaced values are those within one std away from the optimal values for a given fixed $k$.

**Amazon, DBLP.** We first preprocess these networks by following a standard approach described in Section A.2 of the Supplement. As opposed to the networks in the previous two categories, the information in the vertex degrees is extremely predictive of the community membership for this category. Figure 3 shows the predictiveness based on one-step LPs and DNLPs for these two networks. As may be seen, degree normalization may actually hurt the predictive performance of LPs for these two networks. This observation coincides with the finding in [50]. Hence, for this case, we do not perform degree normalization. As recommended in Section 4.2, the weights are chosen as $\gamma_k = \frac{\theta^k}{(\theta^k + \phi)^2}$, where $\theta, \phi$ are parameters to be tuned. The value of $\phi$ typically depends on how informative the degree of a vertex is. Here, we simply set $\phi = \theta^{10}$ which makes $\gamma_k$ achieve its maximal value for $k = 10$. We also find that for both networks, $\alpha = 0.95$ is a good choice for PPR while for HPR, $h = 10$ and $h = 5$ are adequate for the Amazon and the DBLP network, respectively.

Further results are listed in Table 1, indicating that HPR outperforms other PR methods when $k = 5$; HPR is used with parameter $\geq 5$, and the weights for the first 5 steps in HPR increase. This yet again confirms our findings regarding the predictiveness of large-step LPs. For larger $k$, IPR matches the performance of HPR and even outperforms HPR on the DBLP network. Vertex rankings within the communities are available in Section A.2 of the Supplement.

## 6 Discussion and Future Directions

There are many directions that may be pursued in future studies, including:
(1) Our non-asymptotic analysis works for relatively dense graphs for which the minimum degree equals $\bar{d}_{\min} = \omega(\log n)$. A relevant problem is to investigate the behavior of GPR over sparse graphs.
(2) The derived weights ignore the correlations between LPs corresponding to different step-lengths. Characterizing the correlations is a particularly challenging and interesting problem.
(3) Recently, research for network analysis has focused on networks with higher-order structures. PPR and HPR-based methods have been generalized to the higher-order setting [52, 53]. Analysis has shown that these higher-order GPR methods may be used to detect communities of networks that approximates higher-order network (motif/hypergraph) conductance [54, 53]. Related works also showed that PR-based approaches are powerful for practical community detection with higher-order structures [55]. Hence, generalizing our analysis to higher-order structure clustering is another topic for future consideration. A follow-up work on the mean-field analysis of higher-order GPR methods may be found in [56].
(4) Our work provides new insights regarding SE community detection. Re-deriving the non-asymptotic results for other GPR-based applications, including recommender system design and link prediction, is another class of problems of interest. For example, GRP/RW-based approaches are frequently used on commodities-user bipartite graphs of recommender systems. There, one may model the network as a random graph with independent edges that correspond to one-time purchases governed by preference scores of the users. Similarities of vertices can also be characterized by GPRs and used to predict emerging links in networks [4]. In this setting, it is reasonable to assume that the graph is edge-independent but with different edge probabilities. Analyzing how the GPR weights influence the similarity scores to infer edge probabilities may improve the performance of current link prediction methods.

## 7 Acknowledgement

This work was supported by the NSF STC Center for Science of Information at Purdue University. The authors also gratefully acknowledge useful discussions with Prof. David Gleich from Purdue University.

## Footnotes

[1]Pan Li and Eli Chien contribute equally to this work.

[2]In some cases, both $\|x^{(k)}\|_2$ and $\|\bar{x}^{(k)}\|_2$ naturally equal to $o(1)$, which leads to the obvious, yet loose bound $\|x^{(k)} - \bar{x}^{(k)}\|_2 \leq \|x^{(k)}\|_2 + \|\bar{x}^{(k)}\|_2 = o(1)$.

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
