[Supplementary Material]

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

# Supplement

We describe the properties of the datasets used in our evaluations as well as more specific algorithmic implementation steps used in our numerical experiments. We then proceed to provide detailed proofs for the main lemmas and theorems.

## A    Supplementary Information for Experiments

Table 2 describes the properties of the datasets used in our evaluations in more detail.

For all real world networks, we first extract the largest connected component of each network in the preprocessing step. For Citeseer and Cora, we arrive at a networks with $2,120$ and $2,485$ vertices, respectively. Other networks considered are connected and thus used in their original form.

| Name | # Vertices | # Edges | # Communities |
|---|---|---|---|
| Citeseer | 3,233 | 9,464 | 6 |
| Cora | 2,708 | 10,858 | 7 |
| PubMed | 19,717 | 88,676 | 3 |
| Amazon | 334,863 | 925,872 | 151,037 |
| DBLP | 317,080 | 1,049,866 | 13,477 |

Table 2: Description of the analyzed real world networks.

### A.1    Additional Observations for the Citeseer, Cora and PubMed Network Evaluations

Figure 4 (Left) demonstrates that for these three networks, PRs with degree normalization perform better than their unnormalized counterparts. Figure 4 (Right) compares IPR-d with different parameters and demonstrates that the performance in the first few steps offered by IPR with $\theta$ equal to $0.95$ and $0.90$ is significantly better than that of IPR with parameter $0.99$, but that it afterwards remains the same or even degrades with increasing $k$. To better understand this phenomenon, we computed the $\lambda_2$ value of these three networks, which equal to $0.9985$, $0.995$ and $0.9859$, respectively. As predicted in Section 4.2, setting $\theta = \lambda_2$ is helpful for obtaining a stable IPR, while more steps are required to "saturate" the performance. In practice, computing $\lambda_2$ for massive networks is time consuming, so we suggest to conservatively select a large $\theta$ and only focus on the first several steps of the random walk. Note that according to our experiments, the averaged maximal lengths of all shortest paths between unlabeled vertices and the seed sets are as follows: Citeseer, $16.0$; Cora, $11.4$; PubMed: $10.6$. Therefore, the recommended choices for $k$ are $80, 57$ and $53$, respectively.

Figure 4: (Left:) Recalls based on one-step LPs and one-step DNLPs. (Right:) Results over the Citeseer (left), Cora (mid) and PubMed (right) networks. Recalls (mean $\pm$ std) of different PRs vs steps.

### A.2    Additional Observations for the Amazon and DBLP Network Evaluations

Note that the evaluation metric we adopt is adequate for communities of similar sizes, which is not the case for the Amazon and DBLP communities. We hence further restrict our choice of community structures to analyze for these two networks. We perform a preprocessing method used in [14, 50]: We select communities closest in size to $m^{3/4}$, where $m$ is the size of the largest community. This leads to 113 communities with sizes in $[1500, 3000]$ for the Amazon network, and 101 communities with

Figure 5: Averaged recalls for different PRs over the Amazon and the DBLP network with respect to the top-$Q$ vertices, obtained by accumulating the LPs for $k \leq 20$.

sizes in $[650, 1000]$ for the DBLP network. For each test, starting from the seed set, we first perform a 4-step (3-step) breadth-first-search to extract a sub-network of the Amazon (DBLP) network. This allows our methods to work locally, and is similar in strategy to the approach adopted in [57]. The obtained sub-networks have $10k$ - $40k$ vertices and cover, on average, more than 70% of the communities of interest. Note that due to the way the sub-networks are constructed, the averaged maximal lengths of all shortest paths between unlabeled vertices and the seed sets are simple to compute: For Amazon, this number equals $4$; for DBLP, $3$. Therefore, accumulating over the first $15 - 20$ steps of the RW is appropriate. Using the obtained sub-networks, we evaluated different GPR approaches.

Figure 5 further illustrates the rankings of vertices within the predicted communities after accumulating the results of the first 20 LPs. Note that for the Amazon network, all three PRs give similar ranking results while for the DBLP network, IPR with parameter $0.9$ outperforms the other two PRs. Again, according to the results for the DBLP network, PPR performs well when ranking vertices with small budgets $Q$.

# B   Proofs of the Results in Section 3

## B.1   Proof of Lemma 3.1

It is straightforward to see that $\mathbb{E}[|A_{uv} - p_{uv}|^2] = p_{uv}(1 - p_{uv}) \leq p_{uv}$. Hence, $\mathbb{E}[\sum_{u \in V} |A_{uv} - p_{uv}|^2] \leq \bar{d}_v$. Using Bernstein's inequality and $\mathbb{E}[|A_{uv} - p_{uv}|] = 2p_{uv}(1 - p_{uv})$, we obtain

$$\mathbb{P}\left[\sum_{u \in V} |A_{uv} - p_{uv}| < 2 \sum_{u \in V} p_{uv}(1 - p_{uv}) - \frac{\epsilon}{2}\bar{d}_v\right] \leq e^{-\frac{\epsilon^2}{10}\bar{d}_v}. \tag{5}$$

Moreover, as $|d_v - \bar{d}_v| = |\sum_{u \in V}(A_{uv} - p_{uv})|$, using Bernstein's inequality once again, we conclude that

$$\mathbb{P}\left[|d_v - \bar{d}_v| > \frac{\epsilon}{2}\bar{d}_v\right] \leq 2e^{-\frac{\epsilon^2}{10}\bar{d}_v}. \tag{6}$$

Therefore, with probability at least $1 - 3e^{-\frac{\epsilon^2}{10}\bar{d}_v}$, it holds that

$$\|x^{(1)} - \bar{x}^{(1)}\|_1 = \sum_{u \in V} |\frac{A_{uv}}{d_v} - \frac{A_{uv}}{\bar{d}_v} + \frac{A_{uv}}{\bar{d}_v} - \frac{p_{uv}}{\bar{d}_v}|$$

$$\geq \frac{\sum_{u \in V} |A_{uv} - p_{uv}|}{\bar{d}_v} - \frac{|d_v - \bar{d}_v|}{\bar{d}_v}$$

$$\overset{a)}{\geq} \frac{2\sum_{u \in V} p_{uv}(1 - p_{uv}) - \frac{\epsilon}{2}\bar{d}_v - \frac{\epsilon}{2}\bar{d}_v}{\bar{d}_v} \overset{b)}{\geq} 2(1 - \frac{\bar{d}_v}{n}) - \epsilon \overset{c)}{\geq} \epsilon,$$

where $a)$ follows from plugging in (5) and (6) into the underlying expression, $b)$ is a consequence of the fact that $\sum_{u \in V} p_{uv}^2 \leq \frac{(\sum_{u \in V} p_{uv})^2}{n} = \frac{\bar{d}_v^2}{n}$ and $c)$ follows from $\bar{d}_v \leq n(1 - \epsilon)$. As $\bar{d}_v = \omega(1)$, the above probability converges to $1$ as $n \to \infty$.

## B.2 Proof of Lemma 3.2 and Lemma 3.5

Before proving Lemma 3.2 and Lemma 3.5 we introduce some useful notation. For a graph with adjacency matrix $A$ and degree matrix $D$, the Randić matrix is defined as $R = D^{-\frac{1}{2}} A D^{-\frac{1}{2}}$. Denote its mean-field value by $\bar{R} = \bar{D}^{-\frac{1}{2}} \bar{A} \bar{D}^{-\frac{1}{2}}$. It is straightforward to see that the eigenvalues of the RW matrix $W$ of an undirected graph are the same as those of $R$. Furthermore, let $d_N$ be the normalized degree vector of $G$, i.e., $d_{N,v} = d_v / \sum_{u \in V} d_u$, and let $D_N$ be the normalized diagonal degree matrix, i.e., $D_N = (\sum_{u \in V} d_u)^{-1} D$. The spectral norm of a matrix $M$ is denoted by $\|M\|_2$. Throughout the Supplement, we also use $\lesssim$ to indicate that the upper bound ignores multiplicative constants.

**Lemma B.1.** For any edge-independent random graph, if $\bar{d}_{\min} = \omega(\log n)$, there exist a constant $C$ and $b \in \{0.5, 1\}$ such that

$$\max_{v \in V} |\frac{d_v^b}{\bar{d}_v^b} - 1| \lesssim \sqrt{\frac{\log(n)}{\bar{d}_{\min}}} \ w.h.p.$$

*Proof.* Based on the Bernstein's inequality, we have

$$\mathbb{P}\left( \sum_{u \in V} (A_{vu} - p_{vu}) > c \sqrt{\sum_{u \in V} p_{vu} \log n} \right) \leq n^{-c/4}.$$

Note that $\sum_{u \in V} (A_{vu} - p_{vu}) = \sum_{u \in V} A_{vu} - \sum_{u \in V} p_{vu}) = d_v - \bar{d}_v$. By dividing both sides by $\bar{d}_v$ and using the union bound, we have

$$\mathbb{P}\left( \max_{v \in V} |\frac{d_v}{\bar{d}_v} - 1| > \max_{v \in V} c \sqrt{\frac{\log n}{\bar{d}_v}} \right) \leq 2n^{-(c/4-1)}.$$

Moreover, choosing $c > 3$ and observing that for all $v \in V$, $\bar{d}_v - c\sqrt{\bar{d}_v \log n} \leq d_v \leq \bar{d}_v + c\sqrt{\bar{d}_v \log n}$ w.h.p., we have $-\frac{c}{2}\sqrt{\log n} \lesssim d_v^{\frac{1}{2}} - \bar{d}_v^{\frac{1}{2}} \lesssim \frac{c}{2}\sqrt{\log n}$ w.h.p. This proves the claimed result. $\square$

**Lemma B.2.** For any edge-independent random graph model, if $\bar{d}_{\min} = \omega(\log n)$, then

$$\|d_N - \bar{d}_N\|_2 \lesssim \sqrt{\frac{\log n}{n \bar{d}_{\min}}} \ w.h.p.$$

*Proof.* Let $d_N' = (\frac{d_v}{\sum_{u \in V} \bar{d}_u})_{v \in V}$. Then,

$$\|d_N - \bar{d}_N\|_2 \leq \|d_N - d_N'\|_2 + \|d_N' - \bar{d}_N\|_2.$$

We separately establish bounds on the two terms of the sum. First,

$$\|d_N - d_N'\|_2 = (\sum_{v \in V} d_v^2)^{\frac{1}{2}} \frac{|\sum_{u,u' \in V} (A_{uu'} - p_{uu'})|}{\sum_{u \in V} d_u \sum_{u \in V} \bar{d}_u}$$

$$\overset{a)}{\lesssim} (\sum_{v \in V} d_v^2)^{\frac{1}{2}} \frac{\sqrt{\sum_{u \in V} \bar{d}_u \log n}}{\sum_{u \in V} d_u \sum_{u \in V} \bar{d}_u} \overset{b)}{\leq} \sqrt{\frac{\log n}{\sum_{u \in V} \bar{d}_u}} \leq \sqrt{\frac{\log n}{n \bar{d}_{\min}}} \ w.h.p.,$$

where $a)$ follows from Bernstein's inequality and $b)$ from Cauchy's inequality. Second,

$$\|d_N' - \bar{d}_N\|_2 = \frac{[\sum_{v \in V} (d_v - \bar{d}_v)^2]^{\frac{1}{2}}}{\sum_{u \in V} \bar{d}_u} \overset{a)}{\lesssim} \frac{(\sum_{v \in V} \bar{d}_v \log n)^{\frac{1}{2}}}{\sum_{u \in V} \bar{d}_u} \leq \sqrt{\frac{\log n}{n \bar{d}_{\min}}}$$

w.h.p., where $a)$ is a consequence of Lemma B.1. Combining the two above results establishes the claim. $\square$

**Lemma B.3.** Let $d_N^{\frac{1}{2}}$ be the vector obtained by taking the square root of the elements in the vector $d_N$. If $\bar{d}_{\min} = \omega(\log n)$, then

$$\|d_N^{\frac{1}{2}}(d_N^{\frac{1}{2}})^T - \bar{d}_N^{\frac{1}{2}}(\bar{d}_N^{\frac{1}{2}})^T\|_2 \lesssim \sqrt{\frac{\log(n)}{\bar{d}_{\min}}} \; w.h.p.$$

*Proof.* First, we have

$$\|d_N^{\frac{1}{2}}(d_N^{\frac{1}{2}})^T - \bar{d}_N^{\frac{1}{2}}(\bar{d}_N^{\frac{1}{2}})^T\|_2 \leq \sqrt{trace[(d_N^{\frac{1}{2}}(d_N^{\frac{1}{2}})^T - \bar{d}_N^{\frac{1}{2}}(\bar{d}_N^{\frac{1}{2}})^T)^2]}$$

$$= \sqrt{2 - 2[(d_N^{\frac{1}{2}})^T \bar{d}_N^{\frac{1}{2}}]^2} \overset{a)}{\leq} \sqrt{4(1 - (d_N^{\frac{1}{2}})^T \bar{d}_N^{\frac{1}{2}})} = 2\|d_N^{\frac{1}{2}} - \bar{d}_N^{\frac{1}{2}}\|_2,$$

where $a)$ follows from $1 + (d_N^{\frac{1}{2}})^T \bar{d}_N^{\frac{1}{2}} \leq 2$. To bound $\|d_N^{\frac{1}{2}} - \bar{d}_N^{\frac{1}{2}}\|_2$, let $d_N^{'\frac{1}{2}} = (\frac{d_v^{\frac{1}{2}}}{\sqrt{\sum_{u \in V} \bar{d}_u}})_{v \in V}$. Then,

$$\|d_N^{\frac{1}{2}} - \bar{d}_N^{\frac{1}{2}}\|_2 \leq \|d_N^{\frac{1}{2}} - d_N^{'\frac{1}{2}}\|_2 + \|d_N^{'\frac{1}{2}} - \bar{d}_N^{\frac{1}{2}}\|_2.$$

We establish bounds for the two terms as:

$$\|d_N^{\frac{1}{2}} - d_N^{'\frac{1}{2}}\|_2 = (\sum_{v \in V} d_v)^{\frac{1}{2}} |\frac{1}{\sqrt{\sum_{u \in V} d_u}} - \frac{1}{\sqrt{\sum_{u \in V} \bar{d}_u}}|$$

$$\overset{a)}{\lesssim} (\sum_{v \in V} d_v)^{\frac{1}{2}} \frac{|\sqrt{\sum_{u \in V} \bar{d}_u} - \sqrt{\sum_{u \in V} \bar{d}_u \pm c\sqrt{\sum_{u \in V} \bar{d}_u \log n}}|}{\sqrt{\sum_{u \in V} d_u \sum_{u \in V} \bar{d}_u}}$$

$$\overset{b)}{\lesssim} (\sum_{v \in V} d_v)^{\frac{1}{2}} \frac{\sqrt{\log n}}{\sqrt{\sum_{u \in V} d_u \sum_{u \in V} \bar{d}_u}} = \sqrt{\frac{\log n}{\sum_{u \in V} \bar{d}_u}} \leq \sqrt{\frac{\log n}{n \bar{d}_{\min}}},$$

where $a)$ is a consequence of Bernstein's inequality while $b)$ only includes the dominant term, and

$$\|d_N^{'\frac{1}{2}} - \bar{d}_N^{\frac{1}{2}}\|_2 \leq \frac{[\sum_{v \in V} (\sqrt{d_v} - \sqrt{\bar{d}_v})^2]^{\frac{1}{2}}}{\sqrt{\sum_{u \in V} \bar{d}_u}}$$

$$\overset{a)}{\lesssim} \frac{[\sum_{v \in V} (\sqrt{\bar{d}_v \pm \sqrt{\bar{d}_v \log n}} - \sqrt{\bar{d}_v})^2]^{\frac{1}{2}}}{\sqrt{\sum_{u \in V} \bar{d}_u}}$$

$$\overset{b)}{\lesssim} \frac{(\sum_{v \in V} \log n)^{\frac{1}{2}}}{\sqrt{\sum_{u \in V} \bar{d}_u}} \leq \sqrt{\frac{\log n}{\bar{d}_{\min}}},$$

where $a)$ is again a consequence of Bernstein's inequality while $b)$ only includes the dominant term. Hence, we have $\|d_N^{\frac{1}{2}} - \bar{d}_N^{\frac{1}{2}}\|_2 \lesssim \sqrt{\frac{\log n}{\bar{d}_{\min}}} \; w.h.p.$, as claimed. $\qquad\square$

The next result can be obtained by invoking Theorem 2 of [58].

**Lemma B.4.** If $\bar{d}_{\min} = \omega(\log n)$, then there exists some constant $C_4$ such that

$$\|R - \bar{R}\|_2 \leq C_4 \sqrt{\frac{\log n}{\bar{d}_{\min}}} \; w.h.p.$$

Moreover, recall that $\bar{\lambda} = \max\{|\bar{\lambda}_2|, |\bar{\lambda}_n|\}$ and let $\lambda = \max\{|\lambda_2|, |\lambda_n|, |\bar{\lambda}_2|, |\bar{\lambda}_n|\}$. Using Weyl's Theorem [59], one has $\lambda \leq \bar{\lambda} + C_4 \sqrt{\frac{\log n}{\bar{d}_{\min}}} \; w.h.p.$

Now, let us turn our attention to proving Lemma 3.2. First, let $W_N = W - d_N \mathbf{1}^T$ and $\bar{W}_N = \bar{W} - \bar{d}_N \mathbf{1}^T$. It is easy to check that $W_N(x - d_N) = W_N x = Wx - d_N$. Moreover, as $W_N d_N = d_N$, we have

$$W_N^k x^{(0)} - d_N = W_N^k(x^{(0)} - d_N) = W^k x^{(0)} - d_N = x^{(k)} - d_N. \tag{7}$$

Let $d_N^{\frac{1}{2}}$ be the vector obtained by taking the square root of the elements in the vector $d_N$ and define $R_N = R - d_N^{\frac{1}{2}}(d_N^{\frac{1}{2}})^T$. Then, $W_N = D^{\frac{1}{2}} R_N D^{-\frac{1}{2}}$. Note that $R_N$ essentially equals the Randić matrix with its first principle component removed. Then,

$$\|R_N^i\|_2 \leq \max\{|\lambda_2|, |\lambda_n|\}^i. \tag{8}$$

Furthermore,

$$\|W_N^k x^{(0)} - \bar{W}_N^k x^{(0)}\|_2/\|x^{(0)}\|_2$$
$$\leq \|W_N^k - \bar{W}_N^k\|_2 = \|D^{\frac{1}{2}} R_N^k D^{-\frac{1}{2}} - \bar{D}^{\frac{1}{2}} \bar{R}_N^k \bar{D}^{-\frac{1}{2}}\|_2$$
$$\overset{a)}{\leq} \|(D^{\frac{1}{2}} - \bar{D}^{\frac{1}{2}})R_N^k D^{-\frac{1}{2}}\|_2 + \|D^{\frac{1}{2}} R_N^k (D^{-\frac{1}{2}} - \bar{D}^{-\frac{1}{2}})\|_2 + \sum_{i=0}^{k-1} \|\bar{D}^{\frac{1}{2}} R_N^{k-1-i}(R_N - \bar{R}_N)\bar{R}_N^i \bar{D}^{-\frac{1}{2}}\|_2$$
$$\overset{b)}{\lesssim} \sqrt{\frac{\bar{d}_{\max}}{\bar{d}_{\min}}} \lambda^{k-1} (2\|I - D^{\frac{1}{2}}\bar{D}^{-\frac{1}{2}}\|_2 + k\|R_N - \bar{R}_N\|_2)$$
$$\leq \sqrt{\frac{\bar{d}_{\max}}{\bar{d}_{\min}}} \lambda^{k-1} (2\|I - D^{\frac{1}{2}}\bar{D}^{-\frac{1}{2}}\|_2 + k\|d_N^{\frac{1}{2}}(d_N^{\frac{1}{2}})^T - \bar{d}_N^{\frac{1}{2}}(\bar{d}_N^{\frac{1}{2}})^T\|_2 + k\|R - \bar{R}\|_2)$$
$$\overset{c)}{\lesssim} k\lambda^{k-1} \sqrt{\frac{\bar{d}_{\max}\log n}{\bar{d}_{\min}^2}} \leq k\left(\bar{\lambda} + C_4\sqrt{\frac{\log n}{\bar{d}_{\min}}}\right)^{k-1}\sqrt{\frac{\bar{d}_{\max}\log n}{\bar{d}_{\min}^2}} \tag{9}$$

where $a)$ is a consequence of the triangle inequality, $b)$ is based on inequality (8) and Lemma B.1 that guarantees $d_{\max} \lesssim \bar{d}_{\max}$ and $\bar{d}_{\min} \lesssim d_{\min}$, and $c)$ follows from Lemma B.1, Lemma B.3 and Lemma B.4. To prove the first inequality, we observe that

$$\|x^{(k)} - \bar{x}^{(k)}\|_2/\|x^{(0)}\|_2$$
$$\leq \|d_N - \bar{d}_N\|_2/\|x^{(0)}\|_2 + \|(x^{(k)} - d_N) - (\bar{x}^{(k)} - \bar{d}_N)\|_2/\|x^{(0)}\|_2$$
$$\overset{a)}{=} \|d_N - \bar{d}_N\|_2/\|x^{(0)}\|_2 + \|W_N^k x^{(0)} - d_N - (\bar{W}_N^k x^{(0)} - \bar{d}_N)\|_2/\|x^{(0)}\|_2$$
$$\leq 2\|d_N - \bar{d}_N\|_2/\|x^{(0)}\|_2 + \|W_N^k x^{(0)} - \bar{W}_N^k x^{(0)}\|_2/\|x^{(0)}\|_2$$
$$\overset{b)}{\lesssim} \sqrt{\frac{\log n}{n\bar{d}_{\min}}} \frac{1}{\|x^{(0)}\|_2} + k\left(\bar{\lambda} + C_4\sqrt{\frac{\log n}{\bar{d}_{\min}}}\right)^{k-1}\sqrt{\frac{\bar{d}_{\max}\log n}{\bar{d}_{\min}^2}},$$

where $a)$ is a consequence of (7) and $b)$ follows based on Lemma B.2 and the inequality (9). This proves the first inequality in Lemma 3.2. Since we have $\|pr(\gamma, x^{(0)}) - \bar{pr}(\gamma, x^{(0)})\|_2 \leq \sum_{k=0}^{\infty} \gamma_k \|x^{(k)} - \bar{x}^{(k)}\|_2$, the bound for $\|pr(\gamma, x^{(0)}) - \bar{pr}(\gamma, x^{(0)})\|_2$ may be derived by using an analysis similar to the one applied to $\|x^{(k)} - \bar{x}^{(k)}\|_2$.

Lemma 3.5 may be established in a similar manner. However, due to degree normalization, one can remove the dependence on $\|d_N - \bar{d}_N\|_2$. Recall once again the definition of the DNLPs $z^{(k)}$ and the normalized degree matrix $D_N$, which allow us to write $z^{(k)} = D_N^{-1}x^{(k)}$. The LHS of the result in

Lemma 3.5 may be rewritten as

$$
\|D_N^{-1}x^{(k)} - \bar{D}_N^{-1}\bar{x}^{(k)}\|_2 / \|\bar{D}_N^{-1}x^{(0)}\|_2
$$
$$
=\|D_N^{-1}(x^{(k)} - d_N) - \bar{D}_N^{-1}(\bar{x}^{(k)} - \bar{d}_N)\|_2 / \|\bar{D}_N^{-1}x^{(0)}\|_2
$$
$$
=\|D_N^{-1}(W_N^k x^{(0)} - d_N) - \bar{D}_N^{-1}(\bar{W}_N^k \bar{x}^{(0)} - \bar{d}_N)\|_2 / \|\bar{D}_N^{-1}x^{(0)}\|_2
$$
$$
\overset{a)}{=}\|D_N^{-1}W_N^k x^{(0)} - \bar{D}_N^{-1}\bar{W}_N^k \bar{x}^{(0)}\|_2 / \|\bar{D}_N^{-1}x^{(0)}\|_2
$$
$$
=\|\frac{\sum_{v\in V} d_v}{\sum_{v\in V} \bar{d}_v} D^{-\frac{1}{2}} R_N^k D^{-\frac{1}{2}} \bar{D} - \bar{D}^{-\frac{1}{2}} \bar{R}_N^k \bar{D}^{\frac{1}{2}}\|_2
$$
$$
\leq \left|\frac{\sum_{v\in V} d_v}{\sum_{v\in V} \bar{d}_v} - 1\right| \|D^{-\frac{1}{2}} R_N^k D^{-\frac{1}{2}} \bar{D}\|_2 + \|D^{-\frac{1}{2}} - \bar{D}^{-\frac{1}{2}}\|_2 \|R_N^k\|_2 \|D^{-\frac{1}{2}} \bar{D}\|_2
$$
$$
+ \|\bar{D}^{-\frac{1}{2}}\|_2 \|R_N^k\|_2 \|D^{-\frac{1}{2}} \bar{D} - \bar{D}^{\frac{1}{2}}\|_2 + \|\bar{D}^{-\frac{1}{2}}(R_N^k - \bar{R}_N^k)\bar{D}^{\frac{1}{2}}\|_2
$$
$$
\overset{b)}{\leq} \sqrt{\frac{\bar{d}_{\max}}{\bar{d}_{\min}}} \lambda^{k-1} 3\|I - \bar{D}^{-\frac{1}{2}} D^{\frac{1}{2}}\|_2 + \sum_{i=0}^{k-1} \|\bar{D}^{\frac{1}{2}} R_N^{k-1-i}(R_N - \bar{R}_N)\bar{R}_N^i \bar{D}^{-\frac{1}{2}}\|_2
$$
$$
\overset{c)}{\lesssim} k\left(\bar{\lambda} + C_4\sqrt{\frac{\log n}{\bar{d}_{\min}}}\right)^{k-1} \frac{\sqrt{\bar{d}_{\max} \log n}}{\bar{d}_{\min}},
$$

where $a)$ follows from (7), $b)$ is a consequence of Lemma B.1 and c) is based on the same arguments used to establish (9).

## B.3    Proof of Theorem 3.3

First, we note that $\|x^{(k)} - \bar{x}^{(k)}\|_1 \leq \sqrt{n}\|x^{(k)} - \bar{x}^{(k)}\|_2$. Based on Lemma 3.2, it is easy to see that if $\|x^{(0)}\|_2 = O(\frac{1}{\sqrt{n}})$, $\frac{\bar{d}_{\max}\log n}{\bar{d}_{\min}^2} = o(1)$, one has $\|x^{(k^{(n)})} - \bar{x}^{(k^{(n)})}\|_1 = o(1)$ $w.h.p.$

If $\bar{\lambda} < 1 - c$, $\frac{c}{3} > C_4\sqrt{\frac{\log n}{\bar{d}_{\min}}}$ and $k^{(n)} \geq \frac{\log n + \log \frac{\bar{d}_{\max}}{\bar{d}_{\min}}}{c}$, then for large enough $n$,

$$
k^{(n)}\left(\bar{\lambda} + C_4\sqrt{\frac{\log n}{\bar{d}_{\min}}}\right)^{k^{(n)}-1} \leq k^{(n)}\left(1 - \frac{2c}{3}\right)^{k^{(n)}-1} \leq \frac{1}{c}\left(\frac{n\bar{d}_{\max}}{\bar{d}_{\min}}\right)^{-\frac{1}{2}}.
$$

In this case, we also have $\|x^{(k^{(n)})} - \bar{x}^{(k^{(n)})}\|_1 = o(1)$ $w.h.p.$

## B.4    Proof of Theorem 3.4

The result in 1) is a consequence of Lemma 3.2,

$$
\|pr(\gamma^{(n)}, x^{(0)}) - \bar{pr}(\gamma^{(n)}, x^{(0)})\|_1 \leq \sqrt{n}\|pr(\gamma^{(n)}, x^{(0)}) - \bar{pr}(\gamma^{(n)}, x^{(0)})\|_2.
$$

Suppose that $\sum_{k\geq 1}\gamma_k^{(n)} \leq B$ for large enough $n$. Then,

$$
\sum_{k\geq 1}\gamma_k k\left(\bar{\lambda} + C_4\sqrt{\frac{\log n}{\bar{d}_{\min}}}\right)^{k-1} \leq B\max_{k\geq 1} k\left(1 - \frac{c}{2}\right)^{k-1} \leq \frac{B}{\frac{2-c}{2}\ln\frac{2}{2-c}} = O(1).
$$

As $\gamma_0^{(n)} \geq C_5\sum_k \gamma_k^{(n)}$, we have $\|\bar{pr}(\gamma^{(n)}, x^{(0)})\|_2 \geq C_5\|x^{(0)}\|_2$. Hence,

$$
\frac{\|pr(\gamma^{(n)}, x^{(0)}) - \bar{pr}(\gamma^{(n)}, x^{(0)})\|_2}{\|\bar{pr}(\gamma^{(n)}, x^{(0)})\|_2} \leq C_5\frac{\|pr(\gamma^{(n)}, x^{(0)}) - \bar{pr}(\gamma^{(n)}, x^{(0)})\|_2}{\|x^{(0)}\|_2}
$$

Lemma 3.2 ensures that the result in 2) is met. The result in 3) is again a consequence of Lemma 3.2 because

$$
\|pr(\gamma^{(n)}, x^{(0)}) - \bar{pr}(\gamma^{(n)}, x^{(0)})\|_1 \leq \sqrt{n}\|pr(\gamma^{(n)}, x^{(0)}) - \bar{pr}(\gamma^{(n)}, x^{(0)})\|_2,
$$

and for large enough $n$, $\bar{\lambda} + C_4\sqrt{\log n / \bar{d}_{\min}} \leq \bar{\lambda} + C_6$.

## C   Derivation of the Means

For notational simplicity, we let $\beta_1 = \frac{n_1 p_1}{n_1 p_1 + n_0 q}$ and $\beta_0 = \frac{n_0 p_0}{n_1 q + n_0 p_0}$. Furthermore, we use $P_i^{(k)} = \sum_{v \in C_i} \bar{x}_v^{(k)}$, $i \in \{0, 1\}$ to denote the sum of $k$-step LPs within the block $C_i$. Due to the symmetry, $\{P_i^{(k)}\}_{i \in \{0,1\}}$ may be obtained from the following recursion, with initial conditions $[P_1^{(0)}, P_0^{(0)}] = [1, 0]$:

$$\begin{bmatrix} P_1^{(k)} \\ P_0^{(k)} \end{bmatrix} = W' \begin{bmatrix} P_1^{(k-1)} \\ P_0^{(k-1)} \end{bmatrix}, \text{ where } W' = \begin{bmatrix} \beta_1 & 1 - \beta_0 \\ 1 - \beta_1 & \beta_0 \end{bmatrix}.$$

Consequently, $\mu_1^{(k)} = \bar{z}_v^{(k)} = \frac{\bar{x}_v^{(k)}}{\bar{d}_v} = \frac{P_1^{(k)}/n_1}{n_1 p_1 + n_0 q}$ and $\mu_0^{(k)} = \frac{P_0^{(k)}/n_0}{n_0 p_0 + n_1 q}$. It is straightforward to show that the matrix $W'$ has eigenvalues 1 and $\beta_1 + \beta_0 - 1$, and that $\beta_1 + \beta_0 - 1$ equals $\bar{\lambda}_2$ of the mean-field random walk matrix $\bar{W}$. Combining $\mu_1^{(k)}$, $\mu_0^{(k)}$ and $\bar{\lambda}_2 = \beta_1 + \beta_0 - 1$, we arrive at the result of equation (4).