[Reviews · NeurIPS 2019]

Reviewer 1



[Originality] Overall, this paper is well written. The first contribution regarding the convergence property of GPR is an improvement over the work of Avrachenkov et al. [23], where the authors drop or weaken some of the conditions. In the second contribution, the authors consider seed-expansion community detection using GPRs, as an improvement over Kloumann et al. [20] who consider the mean-field LPs and find the optimal value of \alpha for PPR. The authors also try to incorporate the variance as well as the mean, resulting in the use of pseudo Fisher's linear discriminant, and the best weighting under SBM is explored. Empirically observing the asymptotic behavior of the variance, the authors propose to use a weighting that increases as k grows, unlike PPR and HPR. [Quality & Clarity] * Although theoretical results including Lemma 3.2, Theorems 3.3 and 3.4 are quite condensed, the implications made from them seem to help the reader understand the contents. * Characterization of the variances is reasonably described; theoretical upper bounds do not significantly differ from the empirical observations. * Some experimental results on algorithm comparison are not much convincing. On Citeseer, Cora, and PubMed, the authors compare IPR (\theta=0.99) to PPR (\alpha=0.95) and HPR (h=10); here the result for PPR (\alpha=0.99) is missing. Clearly, the examined PPR's weight decays much faster than the examined IPR's weight, e.g., 0.95^50=0.08 and 0.99=0.6, and thus, this PPR would not provide a significant change for large k (>50). Also, if my understanding is correct, PPR and IPR demonstrate almost identical performance when \theta and \alpha are nearly equal to 1, and thus, I suspect that IPR (\alpha=0.99) is in fact close to PPR (\theta=0.99). [Minor issues] * Line 136 is confusing: Does the inequality mean that "$\bar{d}_v = \omega(1)$ and $\bar{d}_v \leq (1-\epsilon)n$"? * The small figure embedded in Figure 2 is too small to read. * The markers in Figure 3 are too small. ==== UPDATE ==== Thank you for providing the feedback. The experimental comparison between IPR (0.99) and PPR (0.99) is pretty convincing; it sufficiently demonstrates the empirical difference between them. Also, the feedback clarifies the theoretical insights behind IPR again.

Reviewer 2



1. originality The theoretical results seem to be an improvement over existing results, however, the assumption still seems strong to me. For example, it requires log n * d_max = o(d^2_min) and d_min = w(log n) which does not hold if d_max=log^2 n and d_min = log_n. It means the result only apply on dense graph and not allowing heterogeneous degrees 2. quality The submission seems technically sound, but I did not check the detailed proofs 3. clarity The paper is clearly written, however, a written algorithm (in supplementary if there is no enough space in main paper) on the proposed seed expansion method will make the paper easier to be referenced 4. significance This paper could help practitioners to better understand pagerank, the concentration result can inspire future work which may relax the assumptions

Reviewer 3



This paper is well written with theoretical and methodological improvements on GPR. (1) They derived non-asymptotic bounds for LPs and GPRs, which helps understand previous GPR-based methods. They also generalize the analysis of standard PR methods with fewer assumptions. (2) The proposed IPR method, where they take into account the variance of LPs when selecting weights in the first several steps of RW. They showed IPR has better performance for SE community detection, by simulations and real data on networks with different settings.

[Author Response · NeurIPS 2019]

We thank the reviewers for their time, valuable feedback, and recommendations for improving the manuscript. All the
reviewers seem to agree that our contributions are valid and interesting. *In support of this assessment, we would like to*
*reiterate that IPR is a completely novel PR method for seed-set expansion that defies and disproves the validity of*
*common methods that use decreasing weights for landing probabilities [20].* In our subsequent response, we focus on
further highlighting the differences between IPR and PPR (**Rev1**) and IPR and spectral clustering (**Rev2**). We also
discuss a condition in our theoretical results questioned by **Rev2**, and address **Rev3**'s concern about future work.
**\*IPR vs PPR with parameter 0.99. (Rev1)** The reviewer's intuition that PPR with a parameter close to 1 has
a performance similar to IPR is correct. However, this special case does not imply that IPR is merely a simple
modification of PPR. **Rev1** seemed to overlook the key new insight motivating the IPR method, demonstrated both by
new theoretical results and experiments described in the manuscript: The discriminative power of large-step LPs does
not decrease or decrease as fast as previously expected based on a mean-field analysis alone [20]. Our finding has truly
far-reaching consequences as it shows that near-optimal seed-expansion requires choosing geometrically increasing
rather than geometrically decreasing weights; similar results may be derived for recommender systems/link prediction.
Hence, IPR and PPR lead to fundamentally different implications instead of merely experimental performance.
As requested by the reviewer, we evaluated the performance of PPR with parameter $0.99$ and summarized the results in
the figures/table below. The experimental condition are the same as described in the manuscript. The performance gap
between PPR $0.99$ and IPR is smaller than the gap between PPR $0.95$ and IPR. However, the gap is still significant
(more than one standard deviation) for real world networks and even significantly more so for SBMs.
19

Figure 1: (Left): Recalls (mean $\pm$ std) for different PRs over SBMs with parameters $(500, 0.05, 500, 0.05, q)$, $q = 0.02$; (Right): Recalls (mean $\pm$ std) of different PRs over the Citeseer, Cora and PubMed networks (from left to right).
20

**\*Why not use spectral clustering? (Rev2)** Spec-
tral clustering and seed-expansion approaches have
fundamentally different objectives. Spectral clus-
tering is used to find *all* communities in networks
and is therefore a global clustering algorithm that
does not scale for large networks. Seed-expansion

| Step size $k$ | 5 | 10 | 15 | 20 | 5 | 10 | 15 | 20 |
|---|---|---|---|---|---|---|---|---|
| | Amazon (std: $\pm 0.12$) | | | | DBLP (std: $\pm 0.09$) | | | |
| IPR0.99 | 46.63 | 48.03 | **48.43** | **48.53** | 27.58 | 28.78 | **29.18** | **29.27** |
| IPR0.90 | 46.67 | 48.08 | **48.45** | **48.53** | 27.64 | **29.14** | **29.26** | **29.32** |
| PPR0.95 | 46.57 | 47.92 | 48.30 | 48.43 | 27.46 | 28.49 | 28.90 | 29.06 |
| PPR0.99 | 46.59 | 47.94 | 48.34 | 48.45 | 27.51 | 28.58 | 29.00 | 29.14 |

approaches are local, and their complexity is dictated by the size of the community we are interested in [25]. Fur-
thermore, in its classic form, spectral clustering partitions network nodes and may not be used to obtain overlapping
communities; seed-expansion approaches can easily detect overlapping communities, but to keep our exposition focused
we only analyzed the non-overlapping setting also pursued in [20, 24]. In conclusion, spectral clustering is not expected
to scale and perform well on networks with overlapping communities such as the Amazon and DBLP networks. All
forms of community detection, including classical spectral clustering and PR methods, require knowledge of some input
parameters. In the former case, one needs to know the number of communities, while for PRs one needs to select the
parameters $\alpha$ (for PPR), $h$ (for HPR) or $\theta$ (for IPR). For IPRs, the choice of $\theta$ (and not the parameter $q$ that is only used
in the SBM) also allows for adapting IPRs to different networks and different tasks and is in general easy to estimate.
Even when the parameter is estimated imprecisely, it does not influence the bulk performance gain of IPR compared to
PPR, as the crucial point is that the parameter is used to control the increase (rather than decrease) in the weights.
**\*Conditions used to establish the theoretical results. (Rev2)** Please note that our results already improved the condi-
tion $d_{\max}/d_{\min} = \Theta(1)$ in Avrachenkov et al. [23,24] to $\log n * d_{\max} = o(d_{\min}^2)$. This improvement essentially allows
for much larger heterogeneity of degrees. A significant contribution of our work is the first known characterization of
the variance of LPs under assumptions weaker than any other previously reported ones.
**\*Future directions and improvement. (Rev3)** In the Supplement, we listed four future research directions regarding
how to further improve the GPR framework for seed-expansion community detection, especially in settings for which
IPR may not be optimal. It would be of interest to characterize the correlation between LPs of different steps, as the
correlation between $k$-step and $k+1$-step LPs increases with $k$. Correlations may help in identifying the optimal number
of steps of LPs to accumulate. Moreover, our current theoretical analysis requires communities to be non-overlapping
(also used in [20, 24]). Overlapping as well as *sparse* community GPR methods are other interesting new directions.
**\*Minor issues.** In the revision we will clarify the inequality on line 134 (**Rev1**), enlarge the fonts in the figures (**Rev1**),
provide a more detailed algorithm in the Supplement (**Rev2**). Regarding aggregating results of different communities
(**Rev2**), if we understand correctly, this is what we did in the experiments.

[Meta-Review · NeurIPS 2019]

This paper considers different weighting schemes for aggregating landing probabilities with applications to graph clustering. There is consensus that the submission presents some interesting and solid contributions, both theoretically and empirically. The proposed Inverse PageRank is interesting in that the weights increase for the first few landing probability walk lengths (however, this is also true for the heat kernel). A few paper clarification issues outlined by the reviewers should be addressed in a camera version.